# An Overview of the Ferroptosis Hallmarks in Friedreich’s Ataxia

**DOI:** 10.3390/biom10111489

**Published:** 2020-10-28

**Authors:** Riccardo Turchi, Raffaella Faraonio, Daniele Lettieri-Barbato, Katia Aquilano

**Affiliations:** 1Department of Biology, University of Rome Tor Vergata, 00133 Rome, Italy; riccardoturc@gmail.com; 2Department of Molecular Medicine and Medical Biotechnologies, University of Naples Federico II, 80131 Naples, Italy; raffaella.faraonio@unina.it; 3IRCCS Fondazione Santa Lucia, 00179 Rome, Italy

**Keywords:** Iron metabolism, FRDA, ROS, neurodegeneration, lipid metabolism, GPX4, frataxin

## Abstract

Background: Friedreich’s ataxia (FRDA) is a neurodegenerative disease characterized by early mortality due to hypertrophic cardiomyopathy. FRDA is caused by reduced levels of frataxin (FXN), a mitochondrial protein involved in the synthesis of iron-sulphur clusters, leading to iron accumulation at the mitochondrial level, uncontrolled production of reactive oxygen species and lipid peroxidation. These features are also common to ferroptosis, an iron-mediated type of cell death triggered by accumulation of lipoperoxides with distinct morphological and molecular characteristics with respect to other known cell deaths. Scope of review: Even though ferroptosis has been associated with various neurodegenerative diseases including FRDA, the mechanisms leading to disease onset/progression have not been demonstrated yet. We describe the molecular alterations occurring in FRDA that overlap with those characterizing ferroptosis. Major conclusions: The study of ferroptotic pathways is necessary for the understanding of FRDA pathogenesis, and anti-ferroptotic drugs could be envisaged as therapeutic strategies to cure FRDA.

## 1. Introduction

FRDA is an inherited autosomal recessive neurodegenerative disorder caused by the expansion of a GAA triplet-repeat sequence within the first intron of *FXN* gene, leading to a decrease of frataxin (FXN) [1], a mitochondrial protein involved in the synthesis of iron-sulphur clusters (Fe-S), which are essential for the activity of mitochondrial respiratory chain complexes I, II, and III, Krebs cycle enzyme aconitase and other mitochondrial enzymes [2]. Moreover, FXN is important for cellular iron homeostasis, although its role in this process has not been fully clarified yet. FRDA is characterized by early mortality due to hypertrophic cardiomyopathy. Furthermore, there is a tendency to develop type 2 diabetes in 10% of cases, due to dysfunction of pancreatic β-cells [2] and possibly to the alteration of brown adipose tissue function [3], a mitochondria-enriched tissue, which has been now ascertained to exert an anti-diabetic activity [4].

Mitochondrial energy imbalance, accumulation of mitochondrial iron, uncontrolled production of reactive oxygen species (ROS) and increased lipid peroxidation have been implicated in the pathogenesis of the disorder [5]. Such events resemble to those occurring in a new type of cell death firstly described in 2012 by Dixon et al., that is, ferroptosis [6]. Ferroptosis has peculiar and distinct characteristics compared to other forms of cell deaths. Morphologically, ferroptosis occurs with a decrease in mitochondrial mass and cristae, without the occurrence of nuclear morphological changes and DNA fragmentation [6]. Chromatin condensation, apoptotic body formation, cell shrinkage and caspase activations were not found as in the case of apoptosis; swelling of the cytoplasm and rupture of cell membrane in association with the drop in ATP levels were not found as in the case of necrosis [7]. Furthermore, formation of closed bilayer membrane structures as well as inhibition of the mTOR pathway regulating lysosomal activity were not observed as in the case of autophagy [8].

Even though the accumulation of mitochondrial iron, the increase of oxidative stress and lipid peroxidation found in several cell types with FXN deficiency (e.g., neurons from mouse models and fibroblasts from FRDA patients) [9,10,11,12] are also basically the distinctive hallmarks of ferroptosis, a possible link between FRDA and ferroptosis is emerging only recently.

In this review, we highlight the hallmarks of ferroptosis in FRDA and how improving the knowledge of this iron-dependent cell death could be useful to understand the molecular mechanisms of FRDA and envisage novel therapeutic approaches.

## 2. Ferroptosis

In 2003 Dolma and colleagues identified erastin, a compound capable of selectively killing RAS-expressing cancer cells via a peculiar and still undescribed cell death mode [13]. Erastin blocks the Xc-system, preventing cystine cell import, thus depleting intracellular glutathione (GSH), the main non-enzymatic thiol antioxidant. Few years later, it was discovered that iron chelators were able to inhibit this type of cell death [8]. These observations led to the definition of a new iron-mediated cell death, named ferroptosis [6]. The basis of ferroptosis is the uncontrolled production of specific hydroperoxide phospholipids in the presence of free iron without the proper neutralization by the Xc-system/GSH/glutathione peroxidase 4 (GPX4) axis. The Xc-system imports cystine, the oxidized form of cysteine into cells through a 1:1 exchange with glutamate [14]. It belongs to the heterodimer amino acid transporters family and is composed of a heavy (SLC3A2) and a light chain (SLC7A11) that are finely regulated at the transcriptional level. Nuclear factor erythroid 2-related factor 2 (NRF2) positively regulates SLC7A11 [15]; by contrast, P53 downregulates its expression, leading to a decrease of intracellular cysteine and consequent increased susceptibility to ferroptosis [16]. The imported cystine is reduced to cysteine by GSH and/or thioredoxin reductase 1 and this amino acid is in turn used for the synthesis of GSH [17]. GSH can be oxidized to GSSG by GPX4, a selenium peroxidase that reduces hydroxide complexes, including phospholipid hydroperoxides and cholesterol hydroxide in the respective alcohols, thus inhibiting the lipid peroxidation chain reaction. Hence, inhibition of cystine import through erastin influences the activity of GPX4 and increases susceptibility to ferroptosis [18]. By inhibiting the lipid peroxidation chain reactions, GPX4 represents the main regulator of ferroptosis. In line with its protective role against ferroptosis-related lipid peroxidation, chemical inhibition of GPX4 (e.g., RSL3) leads to lipid peroxide accumulation and cell death [19]. Selenocysteine (SeC) is an amino acid essential for GPX4 antioxidant activity and a specific SeC-tRNA is essential for its insertion into the active site of GPX4 [20]. Studies carried out on mice that express a targeted mutation of the active SeC to Cys (GPX4_U46C) have shown that these mutated mice are viable at birth but die before weaning due to degeneration of a specific subgroup of GABA interneurons in the mouse cortex. Homozygous cells expressing GPX4_U46C are more susceptible to peroxide-induced ferroptosis with respect to controls. Selenium is therefore necessary for the correct functionality of GPX4 and for resistance to pro-ferroptotic conditions, virtually in all cells and tissues [21].

Importantly, two recent studies have identified a new mechanism inhibiting ferroptosis involving the Coenzyme Q oxidoreductase FSP1 (ferroptosis suppressor protein 1) that is independent of GSH-GPX4 system [22,23]. In particular, the suppression of ferroptosis by FSP1 is mediated by coenzyme Q10 that in the reduced form traps lipid peroxyl radicals and restrain the propagation of lipid peroxides, whereas FSP1 catalyses the NAD(P)H-dependent regeneration of coenzyme Q10.

### 2.1. Lipid Metabolism in Ferroptosis

Polyunsaturated fatty acids (PUFAs) are highly prone to a non-enzymatic lipid peroxidation that consists in a chain reaction in which the local production of free reactive oxygen-centred radicals (like HO• and HOO•) can initiate oxidation of a large portion of PUFAs, generating lipid radical (L•) species. Internal radical propagation and peroxidation of L• species finally lead to fragmentation with formation of carbonyl compounds like malondialdehyde (MDA), 4-hydroxynonenal (4-HNE), acrolein and other harmful products that can induce structural rearrangements in cell membranes [24,25,26]. Accumulation of transition metals, such as copper and/or iron oxidizes, lipids through the Fenton reaction producing an uncontrolled quantity of ROS, including HO• and HOO• that in turn increase lipid peroxidation [14]. Lipids containing PUFAs are present in the membrane of many specialized cells, such as the central nervous system (CNS) cells, and in many subcellular organelles [27]. Mitochondria are particularly rich in PUFAs whose role is to maintain the functionality of proteins and membrane transporters and to modulate the mitochondrial dynamics [28].

Acyl-CoA synthetase long-chain family member 4 (ACSL4) has been identified as an important player in lipid peroxidation. ACSL4 is one of the enzymes that activate PUFAs during phospholipid synthesis, by esterifying free fatty acids to CoA, in an ATP-dependent manner, with preference towards long-chain PUFAs such as arachidonic acid (AA). ACSL4 is also involved in the biosynthesis of phosphatidylethanolamine (PE), which contains AA. Doll et al. demonstrated that, by enriching cellular membranes with polyunsaturated fatty acids, ACSL4 enhances the sensitivity to ferroptosis [29]. As consequence, genetic or pharmacological inhibition of ACSL4 reduces the accumulation of lipid peroxides in cells, protecting against ferroptosis [30]. Since oxidized AA-PE are linked to ferroptosis, ACSL4 could be a good therapeutic target to contrast ferroptosis, especially in neurodegenerative disorders in which lipid peroxidation plays a fundamental pathogenic role [31].

### 2.2. Iron Metabolism in Ferroptosis

Ferroptosis is also characterized by alteration of iron homeostasis, as it promotes intracellular and intra-mitochondrial iron accumulation [6]. Iron homeostasis is a finely regulated mechanism. Fe^2+^ released from intestinal cells or erythrocyte degradation is oxidized by ceruloplasmin to Fe^3+^ and bound by transferrin (TF) for serum circulation. TF-Fe^3+^ is recognized by the high affinity membrane protein transferrin receptor 1 (TFR1) and the complex is then internalized via clathrin-dependent endocytosis to form endosomes. The internalization of the TFR1-TF-Fe^3+^ complex represents a rate-limiting step of iron import under normal condition and, consistent with its function, TFR1 protein level is regulated by changes in iron status, being up-regulated or down-regulated by iron decrease or iron increase, respectively. Mechanisms regulating TFR1 expression mainly operate at mRNA post-transcriptional levels through the iron-responsive element (IREs)-iron-responsive-proteins (IRPs) system (see below for detail).

In the endosome compartment, low pH promotes transferrin to release Fe^3+^, which is subsequently reduced to Fe^2+^ by the six-transmembrane epithelial antigen of the prostate 3 (Steap3) and then transported into the cytoplasm by the divalent metal carrier 1 (DMT1). It has been recently demonstrated that DMT1 can be directly involved in mitochondrial Fe^2+^ influx being also localized in the mitochondrial outer membrane [32,33].

The iron transferred from endosomes into cytosol becomes then part of the labile iron pool (LIP) in the mitochondria or cytoplasm, bound to low-molecular-weight molecules [34]. Under physiological conditions, LIP is transitory and contains low iron levels that are catalytically reactive and sufficient for metabolic functions; when in excess, iron is stored by ferritin (FT). FT is a hetero-oligomeric protein complex with two different subunits: ferritin heavy chain (FTH), carrying ferroxidase activity involved in iron release, and ferritin light chain (FTL) for iron nucleation. FT is related to homeostasis of LIP by keeping under control the iron release and recycling [35].

The transport of iron out of the cell is dependent on ferroportin (FPN), which is currently the only known iron exporter. This iron transport into the extracellular compartments requires ferroxidase activity, provided by ceruloplasmin multicopper oxidase (Cp), an enzyme containing six-seven copper atoms, that oxidizes extracellular Fe^2+^ to Fe^3+^ prior the release to transferrin. Cp, as well as other multicopper iron oxidases, is essential for iron transportation, due to its molecular structure that prevents ROS formation unlike the spontaneous reaction of iron. Furthermore, multicopper oxidases have a high affinity for oxygen and this significantly reduces the rate of oxidation [36].

Homeostasis of cellular iron is under the control of two iron-regulatory proteins, namely IRP1/ACO1 and IRP2/IREB2 that are RNA-binding proteins [37]. IRPs are activated by iron deficiency and bind iron-responsive element (IRE) present in the 5′ or 3′ untranslated regions (UTRs) of numerous mRNAs encoding proteins for iron metabolism, like TFR1, DMT1, FPN, FTs subunits (FTH1 and FTL). Transcripts harbouring IREs can be modulated by IRPs in translation and/or stability, depending on IRE locations. In particular, mRNAs with IRE in the 5′ UTRs (e.g., FTs, FPN) are suppressed in translation, while transcripts with IREs in the 3′UTRs are increased in stability (e.g., TFR1, DMT1). As an example, in iron-deficient cells, activated IRPs prevent FT mRNA translation and protect TFR1 mRNA from degradation; on the contrary, under iron-sufficient conditions, low IRP activities favour FT synthesis and destabilize mRNA of TFR1, thus they properly balance iron import and store [38].

Iron amount also controls IRP functions at posttranscriptional level: IRP1 loses its RNA-binding activity and IREB2 is degraded by the proteasome system when iron is increased [38,39]. Accordingly, in mice with targeted deletion of IREB2, LaVaute and colleagues have found an increase of FT protein levels co-localizing with high iron content in degenerating neurons [40]. Therefore, it can be speculated that reducing intracellular iron concentrations could be a means to counteract the ferroptotic cascade in diseases. This is also suggested by recent experiments demonstrating that overexpression of the heat shock beta-1 protein (HSPB1) inhibits ferroptosis induced by erastin [41]. Because HSPB1 acts by decreasing endocytosis and recycling of TFR1, another strategy could be to directly silencing, the gene that codes for TFR1 [42]. Free iron is the main cause of the non-enzymatic lipid peroxidation process; for this reason, ferritinophagy, the autophagic process that leads to degradation of iron storage cell proteins, has been widely studied in relation to ferroptosis. In particular, the nuclear receptor coactivator 4 (NCOA4), which mediates the selective autophagic degradation of FT, has been recently shown to contribute to ferroptosis process by increasing iron pool and ROS through Fenton reaction [43].

## 3. Ferroptosis Markers in FRDA

Some distinctive features of ferroptosis have been recognized in many neurodegenerative diseases such as the accumulation of iron and lipid peroxide production, especially in specific regions of the central and peripheral nervous system [44]. In Alzheimer’s disease (AD), the most common form of dementia, characterized by cognitive impairment and memory loss, elevated iron levels have been found in the hippocampus, which is severely impaired in patients with AD. Free iron causes ROS production through Fenton reactions, leading to massive oxidative damage in this brain region [45]. Iron chelators (e.g., Deferoxamine mesylate, DFO) were therefore used to counteract oxidative damage, reducing ferroptosis induction in AD [46]. Same strategy has been used to ameliorate the degeneration of dopaminergic neurons in *substantia nigra* in Parkinson’s disease patients, as in this region abnormal iron accumulation is observed that is high neurotoxic [47]. Amyotrophic lateral sclerosis (ALS) is a neurodegenerative disease characterized by motor neuron dysfunction and spinal cord impairment. Ferroptotic features of dying motor neurons and increase of ferroptotic markers in blood samples have been reported, and anti-ferroptotic drugs, such as edaravone, have been proposed in parallel with other therapeutic approaches to ALS therapy [48,49].

We recently demonstrated that ferroptosis occurs also in FRDA experimental models [3]; however, even though many are the evidence in literature reporting the appearance of ferroptosis markers in FRDA models and patients, to date ferroptosis as a death mechanism in FRDA has been poorly considered. In the following subparagraphs, we will explore in depth ferroptosis hallmarks in FRDA and underline how the study of the main actors of ferroptosis can help to understand the mechanisms of the disease and hypothesize new possible therapeutic strategies.

### 3.1. Lipid Peroxidation in FRDA

FRDA is characterized by altered lipid metabolism that leads to accumulation of intracellular lipids in the form of lipid droplets (LDs). Actually, it has been reported that FRDA patients show accumulation of LDs in fibroblasts [50]. LD accumulation was also observed in FXN deficient cultured rat cardiomyocytes and iPSC-derived FRDA cardiomyocytes [51,52] as well as in heart and brown adipose tissue of FRDA mouse models [3,53]. Notably, FRDA is also characterized by an increase in lipid peroxidation. In particular, high level of the lipid peroxidation product MDA have been observed in the plasma of FRDA patients [12,54]. However, treatment with ibedenone, an analogue of coenzyme Q, although effective in protecting mitochondrial function and improving cardiac activity, was not able to significantly reduce plasma MDA levels in FRDA children [54]. The same trend was observed for other conventional antioxidants, like tocopherols, which are unable to counteract the high propagation speed of lipid oxidation and the consequent oxidative damage. Hence, many researchers attempted at developing alternative strategies. In ROS-induced oxidation of PUFAs, the speed limitation step is the extraction of hydrogen from a bis-allyl site. Cotticelli and colleagues reduced the propagation speed of the oxidative cascade, by the deuteration of the bis-allyl sites of linoleic acid and alpha-linoleic acid, without changing the chemical structure of the fatty acid. They found a reduction in lipid peroxidation in FXN deficient yeasts and murine fibroblasts [55]. Moreover, the inclusion of a portion of deuterated PUFAs (50%) in the total PUFA pool preserved mitochondrial respiratory function and protected myoblast cells from oxidative stress induced by lipid peroxides [56].

These data indicate that lipid peroxidation is a main element in FRDA progression, but the therapies currently tested are not able to decrease the lipid oxidative state. As mentioned before, inhibition of ASCL4 is a possible therapeutic strategy to reduce lipid peroxide production, but currently, no attempts have been made to target this enzyme in preclinical model of FRDA.

### 3.2. Glutathione and Glutathione Peroxidases (GPXs) in FRDA

Currently, the direct role of GPX4 in FRDA has not been deeply investigated. We have recently showed a significant reduction in the protein and mRNA expression of GPX4 and increased amount of lipid peroxidation in stromal vascular cells isolated from adipose depots of a FRDA mouse model both under resting conditions and upon treatment with the inhibitor of GSH biosynthesis, L-buthionine-(S,R)-sulfoximine (BSO) [3], also known to be a ferroptosis inducer [19]. Moreover, we found an impairment of adipocyte differentiation associated with the increase of ferroptosis markers (e.g., downregulation of GPX4 mRNA and protein). To our knowledge, this was the first evidence of the alteration of GPX4 in FRDA.

Currently, 8 GPXs (GPX1–GPX8) have been identified in mammalian tissues, which show different functions and evolutionary origin [57]. GPX1 is expressed in red blood cells and tissues of the liver, lungs and kidneys and is found in the cytosol, nucleus and mitochondria. The antioxidant effects of GPX1 are obtained with the direct reduction of hydrogen peroxide and lipid hydroperoxides [58]. GPX2 is found only in the cytosol and nucleus of the gastrointestinal tract. Due to its specific localization, GPX2 was the first GPX considered a barrier to the absorption of hydrogen peroxide [59]. GPX3 is present in the mitochondria of various organs, such as kidneys, lungs, epididymis, breasts, heart and muscles. Moreover, it is the main antioxidant enzyme in plasma that reduces ROS products during normal metabolism or oxidative damage [60]. GPX4 is present in three forms that are encoded by a single gene. All three forms have the same core sequence but differ in their N-termini and show distinctive tissue expression and subcellular localization. Cytosolic GPX4 has ubiquitous expression and is the predominant isoform in neurons. It has a pivotal role in brain development and function [61]. The other two GPX4 forms, i.e., sperm nuclear GPX4 and mitochondrial GPX4 (mGPX4) are predominantly expressed in testis [62]. The selenium-independent GPX5 is expressed in the epididymis and protects sperm from peroxide-mediated attacks during maturation [63]. GPX6 is expressed in the olfactory epithelium of humans [64]. Another non-selenium glutathione peroxidase, GPX7, which lacks GPX activity, has recently been described as a new phospholipid hydroperoxide GPX [65]. As a non-selenocysteine, GPX8 is a membrane protein, expressed in lung and has been identified as a new member of the GPX family [66].

In a yeast model of FXN deficiency, the GPX activity is five times less in comparison with the total glutathione (GSH + GSSG) in FXN-deficient cells [67]. The imbalance in GSH and GPX activity has been widely demonstrated in fibroblasts of FRDA patients treated with the GSH depleting agent BSO. In particular, FRDA cells have lower viability compared to cells of healthy controls upon BSO treatment [68,69,70]. Tozzi and collaborators have found a higher ratio of superoxide dismutase/GPX1 activity in the blood of FRDA patients compared to controls [71]. Interestingly, treatment with selenium or vitamin E normalizes the antioxidant activity of myocardial GPX1 and slows the progression of cardiomyopathy [72]. In line with these data, treatment with ferroptosis inhibitors (e.g., SRS11-92) also reduces cell death induced by FXN knockdown in healthy human fibroblasts [70]. Furthermore, idebenone, an antioxidant drug localizing within mitochondria, that is considered as a small-molecule GPX mimetic, increases GPX1 activity and ameliorates cardiomyopathy in FRDA patients [68]. These data confirm the central role of GPXs in regulating intracellular redox state and causing cell dysfunction or death by ferroptosis. Despite this, the role of GPX4 in the development of FRDA is not well characterized. Importantly, the deeper investigation of the crosstalk between FXN and GPX4 could increase our knowledge about FXN in modulating intracellular redox homeostasis. Some works corroborate this assumption as an increase in the antioxidant response was found upon FXN overexpression in tumour cell lines, arguing that this protein could also act as a tumour suppressor [73,74].

### 3.3. Iron Dysmetabolism in FRDA

Iron plays an important role in several biological processes, including oxygen transport, DNA synthesis and repair, cellular metabolism and respiratory activity [75,76]. An uncontrolled accumulation of intracellular iron triggers the generation of ROS through Fenton reactions and oxidative stress, thereby damaging lipids, proteins and DNA [77]. The role of FXN in iron metabolism is still unclear and under study. The mitochondrial ferritin (FTMt) is responsible for iron import from the cytosol into the mitochondrion wherein it is stored, thus reducing cytosolic ROS production [78]. Campanella and collaborators found that the overexpression of FTMt in FXN deficient yeasts preserved mitochondrial DNA integrity recovering respiratory capacity. Furthermore, FTMt overexpression elicited cell resistance to oxidative stress induced by H₂O₂ treatment [79]. Few years later the same authors found that upon FTMt overexpression, fibroblasts of FRDA patients exhibited a decreased ROS production and a partial recovery of the Fe-S mitochondrial enzyme activities [80]. These data provide support to the idea that FXN is closely involved in mitochondrial iron binding and detoxification, pointing out FTMt increase as a possible therapeutic approach to counteract FXN deficiency in FRDA.

In line with these findings, the overexpression of the human ferritin light chain (FTL), responsible for intracellular iron storage, increases life span in FXN deficient yeast by preventing oxidative stress and iron accumulation [81]. Abnormal intracellular iron distribution in FRDA was demonstrated in several studies, and accordingly elevated levels of serum TF receptor were found in FRDA patients [82,83]. This is indicative of limited cytoplasmic iron availability as indeed revealed in fibroblasts and lymphoblasts from FRDA patients [84]. Accordingly, cardiac tissues from conditional FXN knockout mice display altered mRNA expression of 18 genes associated with iron metabolism. In particular, the upregulation of TFR1, SEC15L1 and MFRN2 mRNAs and down-regulation of FPN1 mRNA suggest an activated response to cytosolic iron deficiency and correlate with increased mitochondrial import; however, the mRNA levels of various enzymes involved in biosynthetic pathways utilizing mitochondrial iron (heme and iron-sulfur cluster) were down-regulated [85]. In addition, tissues presented an increased mRNA level of Heme Oxygenase 1 (Hmox1), an essential enzyme for heme catabolism that releases biliverdin, carbon monoxide, and ferrous iron, as well as reduced mRNA amount of FTMt. A reduction of heme in heart tissue was also found [85]. This means that although iron is accumulated in the mitochondria, its utilization is defective. Finally, an up-regulation of NCOA4 has been found in primary cells with FXN deficiency, indicating an increase of ferritinophagy and intracellular iron pool [3].

Collectively, these findings point to a significant cellular response to cytosolic iron deficiency in FRDA models and highlight the importance of the mitochondrial iron unbalance in the activation of ferroptotic pathways through Fenton reactions.

### 3.4. NRF2 in FRDA

There are few transcription factors currently reported as negative regulators of ferroptosis [86]. Among these, NRF2 was firstly described as a crucial player in the protection of hepatocellular carcinoma cells (HCC) against ferroptosis. NRF2 inhibits the ferroptotic cascade through up-regulation of the same genes involved in maintenance of iron homeostasis, such as heavy ferritin chain (FTH1) and quinone oxidoreductase (NQO1), and in heme metabolism, such as Hmox1. The knockdown of NRF2, in fact, increases the susceptibility of HCC cells to antitumor action of ferroptosis inducers, like erastin and sorafenib, both in vivo and in vitro [87,88].

The compromise of NRF2 function has been widely demonstrated both in in vivo and in vitro FRDA models [89,90,91]. NRF2 plays a key role in several pathological processes, including inflammatory processes, antioxidant defence, and in improving mitochondrial activity, that are defective in many neurodegenerative pathologies [92,93]. For this reason, several studies have focused to find possible therapeutic approaches that can recover NRF2 activity. Reisman and colleagues tested omaveloxolone (OMOV), a synthetic oleanane triterpenoid that pharmacologically activates NRF2 in monkeys, and found a concomitance increase of NQO1 activity and mitochondrial respiratory capacity [94]. Remarkably, as systemic exposures to OMOV were readily achievable in FRDA patients after oral administration, this drug was suggested to be a powerful tool to restore NRF2 activity in FRDA [94]. Furthermore, the mRNA and protein levels of NRF2 were tested in FXN deficient fibroblasts treated with different redox compounds such as sulforaphane (SFN), dimethyl fumarate (DMF), N-acetylcysteine (NAC), EPI-743, idebenone and OMAV. All these drugs significantly increased the NRF2 mRNA and protein, with NAC and SFN showing the greatest efficacy in recovering NRF2 mRNA to levels comparable to those of healthy controls [95].

An alternative strategy adopted by Ast and colleagues was to grow FXN-null yeasts, nematodes and human cells under hypoxic conditions (O_2_, 1%). They show that Fe-S clusters are stabilized and NRF2 function is similar to control. The reasons why hypoxia, even in the absence of FXN, can directly boost Fe-S cluster function can be ascribed to 3 possible conditions: (1) recovering of the allosteric activation of the ISC complex member/s, perhaps by conformational change/s under low O_2_; (2) limiting the oxidation of ferrous iron, thus increasing the iron availability, and supporting the reduced state of critical cysteine/s that provide sulphur for the cluster; and (3) delaying degradation of some Fe-S clusters by the ubiquitin ligase FBXL5-IRP2 pathway, a direct sensor system of O_2_, as well as iron [96]. Additionally, the lowering of the redox state given by hypoxia, increased the activity of antioxidant factors including NRF2 [96].

### 3.5. P53 in FRDA

P53 is an important tumour suppressor gene, which regulates cell cycle and apoptosis. The role of P53 in ferroptosis is not fully clarified. In cells harbouring a partial deletion of P53 gene, treatments with oxidative stress inducers do not cause cell death. On the contrary, death was committed when P53 was activated, suggesting that P53 may decrease cell antioxidant capacity. For these reasons, P53 is considered a ferroptosis inducing agent [97]. Moreover, the activation of P53 inhibits the transcription of the Xc system subunit SLC7A11 and also affects GPX4 functionality, triggering a reduction in antioxidant potential and increasing the susceptibility to ferroptotic inducers [16].

The role of P53 has also been studied in FRDA, in particular it has been demonstrated that P53 is involved in cell cycle arrest and apoptosis in a neural model with FXN deficiency [98]. P53 is highly expressed in FXN silenced astrocytes and, as consequence, the levels of the downstream cyclin-dependent kinase P21, which promotes cell cycle arrest, are increased [99]. High P53 activity can decrease the antioxidant potential of FXN-deficient cells since, as above stated, it inhibits the transcription of the component of the cystine/glutamate antiporter, SLC7A11 [16,88]; therefore, the suppression of P53 activity in order to limit Xc system inhibition and the consequent GSH lowering could represent a good therapeutic strategy to counteract oxidative stress and likely ferroptosis in FRDA.

## 4. Conclusions

In published literature, the topic of ferroptosis is mainly discussed in the field of cancer. Inducing iron-mediated cell death in cancer cells pharmacologically is one of the possible strategies that are going to be developed to counteract tumour aggressiveness. In this review, we have highlighted that many of the typical markers of ferroptosis are found in FRDA. Despite this, the role of ferroptosis in FRDA pathogenesis and disease progress has received relatively little attention.

The accumulation of iron inside the cell and in cellular compartments, in particular in the mitochondria, and the relative increase in oxidative stress are distinctive elements of both FRDA and ferroptosis. The production of lipid peroxides and consequent increase of their derived products, such as MDA, that have been found increased in the plasma of FRDA patients, are distinctive signs of iron-mediated cell death. The low levels of GSH found in patients and the poor activity of GPXs, in particular GPX4, suggest a strict connection between ferroptosis and FRDA. A schematic representation of the molecular factors altered in FRDA that are relevant to ferroptosis modulation is presented in Figure 1.

There is no effective cure for FRDA, but many studies have proposed the use of antioxidants to counteract oxidative stress damage. Nevertheless, all the drugs currently tested have shown to have no significant benefits in restoring neural, heart and locomotor function. Hypoxia can be a viable alternative to improve oxidative status at the cellular level, even if a clinical procedure has not been still implemented. In conclusion, the molecular dynamics characterizing FRDA can be correlated to ferroptosis; therefore, therapeutic solutions that delay this type of iron-mediated cell death could provide the key to find new possible treatments for FRDA.

## Figures and Tables

**Figure 1 biomolecules-10-01489-f001:**
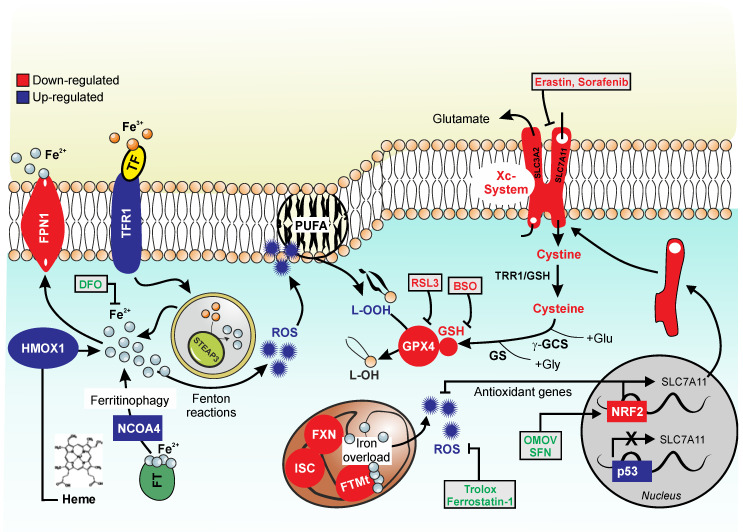
Ferroptotic markers in Friedreich’s ataxia. The figure recapitulates the hallmarks of ferroptosis that are also characteristic of FRDA. Down-regulated factors are represented in red, while up-regulated factors are represented in blue. The figure shows reactive oxygen species (ROS) overproduction due to different causes. FXN-deficiency leads to accumulation of free iron that is not correctly bound by storage systems, such as FTMt and iron-sulfur clusters (ISC) at the mitochondrial level. Moreover, the free iron pool increases due to: augmented activity of heme oxygenase-1 (HMOX-1), which releases free iron from heme degradation; up-regulation of NCOA4, which is involved in ferritinophagy; augmented levels of transferrin receptor 1 (TFR1), which imports iron into the cell; and to insufficient iron export through ferroportin (FPN1). Free iron participates in Fenton reactions and ROS are overproduced causing oxidative stress. Upon FXN deficiency, a reduction in the Xc (SLC7A11/SLC3A2)-system-mediated cystine import also occurs, likely due to increase of P53 and decrease of NRF2 activity that represses and induces SLC7A11 transcription, respectively. This leads to the lowering of intracellular cysteine pool, thus decreasing GSH synthesis and glutathione peroxidase 4 (GPX4) activity. An overproduction of harmful lipid peroxides (LOOH) is elicited, that further contribute to ROS overproduction. Finally, the level of Nuclear factor erythroid 2-related factor 2 (NRF2), the main transcription factor involved in the expression of antioxidant genes with a strong anti-ferroptotic potential, results reduced. Ferroptosis inducers: RSL3, GPX4 inhibitor; BSO (L-buthionine sulfoximine), GSH synthesis inhibitor; erastin and sorafenib, Xc system inhibitors. Ferroptosis inhibitors: Deferoxamine (DFO), iron chelator; Trolox and Ferrostatin 1, ROS inhibitors; SFN (sulforaphane) and OMOV (omaveloxolone), NRF2 inducers.

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
