# Peer review of "An Overview of the Ferroptosis Hallmarks in Friedreich’s Ataxia"

_biomolecules, 2020, doi:10.3390/biom10111489_

Round 1

Reviewer 1 Report

Turchi et al present an interesting review on Friedrich’s ataxia and its common aspects with ferroptosis. More importantly, they highlight several targets in the pathway for possible therapeutic intervention.

Major points

  • The authors mention activation of mTOR pathway associated to increased lysosomal activity (line 68). This sentence is confusing since autophagy activation usually correlates with mTOR inhibition. We suggest editing this sentence for clarity.
  • The authors describe several well-known ferroptosis inducers like inhibitors of cystine import (line 94) and GPX4 inhibitors (line 97) as well as anti-ferroptotic targets like ACSL4. It would be a good addition to include these inhibitors and their targets in figure 1.
  • A couple of recent studies (PMID: 31634900 and PMID: 31634899) have identified a new mechanism of ferroptosis involving FSP1 and coQ10 reduction that can inhibit lipid peroxidation and ferroptosis independently of GPX4 and glutathione. The authors should mention these studies in section 1.
  • In chapter 3, it could be good for the non-expert reader to add a sentence mentioning that besides the increases in lipid peroxidation, there is also an increase in general lipid metabolism and/or lipid droplet content in FXN deficient cells as the authors showed recently.
  • A list of abbreviations used in the manuscript would be helpful.

Author Response

We are thankful to this Reviewer for the careful review of our manuscript and for the suggestions. We have revised the text and the Figure 1 according to the suggestions  as described in the following point-to-point reply.

 Major points

  • The authors mention activation of mTOR pathway associated to increased lysosomal activity (line 68). This sentence is confusing since autophagy activation usually correlates with mTOR inhibition. We suggest editing this sentence for clarity.

OK. We have edited this sentence as requested (see ll. 85-87)

  • The authors describe several well-known ferroptosis inducers like inhibitors of cystine import (line 94) and GPX4 inhibitors (line 97) as well as anti-ferroptotic targets like ACSL4. It would be a good addition to include these inhibitors and their targets in figure 1.

OK. We have included several inducers/inhibitors of ferroptosis in the Figure 1. However, we have not illustrated ACSL4, as in this Figure, to avoid overcrowding, we have only reported ferroptosis factors that have been found altered in FRDA.

  • A couple of recent studies (PMID: 31634900 and PMID: 31634899) have identified a new mechanism of ferroptosis involving FSP1 and coQ10 reduction that can inhibit lipid peroxidation and ferroptosis independently of GPX4 and glutathione. The authors should mention these studies in section 1.

OK. We have added and discussed these important studies (see ll. 125-129 and new refs. 22,23).

  • In chapter 3, it could be good for the non-expert reader to add a sentence mentioning that besides the increases in lipid peroxidation, there is also an increase in general lipid metabolism and/or lipid droplet content in FXN deficient cells as the authors showed recently.

OK. We have briefly described the alteration of lipid metabolism in FRDA as suggested by citing our works (ref. 3) and other papers describing the accumulation of intracellular lipids (new refs. 50-53; ll. 237-242).

  • A list of abbreviations used in the manuscript would be helpful.

OK. We have added the list of abbreviations as suggested after the keywords.

Reviewer 2 Report

Overall, the authors have done a reasonably good job of summarizing the iron-related pathologies associated with FRDA and the potential contributions that ferroptosis may play in disease progression. There are a number of minor edits related to proper/appropriate use of the English language, but these can be easily addressed with the assistance of a consultant.  As such, these changes will not be the focus of this review.

Line 71: In reference to "FRDA cells", is this meant to imply all cells in patients with FRDA, more highly affected cell types, or the cellular impacts of FRDA? Just needs to be be clarified.

Line 73: "main" is redundant when used with "hallmark"

Lines 85-95: The description in the text may be confusing to the casual reader (important since this is a review article), and a small figure or image (if allowable by the journal) would enhance the reader's understanding of the roles of NRF2 and P53 as it relates to SLC7A11 expression. Also, when discussion "expression" in the manuscript, it would be useful for the authors to clearly indicate whether they are referring to "gene" or "mRNA" expression or protein expression (something that may be better referred to as "protein abundance").

Line 126: Please clarify what is meant by "...ACSL4 imposes the sensitivity to ferroptosis". Provide some directionality: does it contribute to enhanced sensitivity?

Line 138: Following reduction by Steap3, discuss the process through which iron exits endosomes (i.e., describe the role of DMT1 in endosomal iron export). Further phenomenologically, presumably under normal (e.g., without FRDA) conditions, transferrin-transferrin receptor mediated uptake would be controlled by cellular iron status.  If uptake is stimulated, under what conditions would you expect iron released from the endosome to be targeted for storage in ferritin?  Perhaps this is a nuanced discussion, but not all cells have the same cellular demands for iron and this may impact the fate of iron once it is released from the endosome.

Line 139: Fe(II) is the preferred iron substrate for ferroportin. As it written, it appears as though the authors are stating that ferrous iron is oxidized prior to export via ferroportin.  If so, what is the oxidase that is involved in the proposed pathway and how does ferroportin export ferric iron?

Line 140: The review paper referenced discusses the role of both IRP1 and IRP2 (or IREBP2) in the control of ferritin protein synthesis (via translational control). Again, perhaps nuanced, but simply using "expression" does not capture the mode of regulation that is conferred through IRP-dependent mechanisms. Further, the authors' focus should not be solely on IRP2 as this is not reflecting the data that is presented in the reference that is provided.

Line 143: The paper by Brewer et al is not appropriately described. Indeed, in this paper, the authors did not inhibit IRP2. Please confirm that this is the reference that was intended to be used.

Line 145: The paper that is referenced did not assess iron abundance or availability as a result of HSPB1-mediated control of TfR. Because iron was not actually assessed, please indicate that this is a speculation, but that it is consistent with the model.  It appears as though the authors are over-interpreting the data that was actually provided in the paper that is referenced.

Line 202: Alteration of GPX4 activity? Or expression? Or protein abundance?  Please clarify.

Line 203: Revise "Actually, previous studies were focused on other GPXs". As written, it is incomplete. Because of the mention of other GPX isoforms, it would be appropriate to provide additional background related to GPX function and tissue-specific expression. Or, alternatively, focus only on GPX4 (though this would be a mistake).

Line 211: Rather than reference a review article, the authors should instead reference the original research article from Helveston et al (1996).

Line 217: Has the role of GPX4 been "poorly" characterized? This implies that the work has been conducted but it has been of questionable quality.  Consider instead...."the role of GPX4.......FRDA is not well characterized".

Line 229: Revise to remove "a light was shed..."

Line 236: Consider using "lend" or "provide support" rather than "give effort".

Lines 245-246: It does not appear that the paper that is referenced actually measured mitochondrial iron utilization. Please revise accordingly. Similarly, although Fe-S cluster assembly and/or heme biosynthesis were not assessed. This reviewer did not evidence that Huang et al demonstrated an increase in free iron and/or cellular damage.

Line 283-284: Provide some specific examples of how hypoxia impacts Fe-S cluster biogenesis. Additionally, "...limiting Fenton reactions" is incomplete.

Line 285: Again, this can be a more nuanced discussion than is presented, but not all Fe-S containing protein exhibit similar levels of sensitivity to oxygen.

Line 299: Remove the hyphen between "demonstrated" and "that".

Lines 302-305: Clearly describe the impact of the reduced expression of the Xc system.

Line 311: Rather than "...very few articles have deeply explored..." consider "the role of ferroptosis in FRDA pathogenesis and disease progress has received relatively little attention" (or little investigation).

Author Response

We thank this Reviewer for the positive evaluation of our manuscript. We have really appreciated the meticulous revision of the text as well as all the suggestions that have highly improved the quality of our manuscript. Following is a point-to-point reply to the Reviewer comments.

Line 71: In reference to "FRDA cells", is this meant to imply all cells in patients with FRDA, more highly affected cell types, or the cellular impacts of FRDA? Just needs to be be clarified.

OK. We have clarified this issue by citing some papers in which affected cell types in FRDA patients or mouse models are reported (see ll. 89-90).

Line 73: "main" is redundant when used with "hallmark"

OK. We have deleted “main”.

Lines 85-95: The description in the text may be confusing to the casual reader (important since this is a review article), and a small figure or image (if allowable by the journal) would enhance the reader's understanding of the roles of NRF2 and P53 as it relates to SLC7A11 expression. Also, when discussion "expression" in the manuscript, it would be useful for the authors to clearly indicate whether they are referring to "gene" or "mRNA" expression or protein expression (something that may be better referred to as "protein abundance").

OK. We have included the mechanisms by which NRF2 and P53 regulates the expression of SLC7A11 in the revised Figure 1. Moreover, we have indicated throughout the manuscript whether we refer to gene or mRNA expression or protein expression.

Line 126: Please clarify what is meant by "...ACSL4 imposes the sensitivity to ferroptosis". Provide some directionality: does it contribute to enhanced sensitivity?

OK. We have clarified this issue (see ll. 150-151)

Line 138: Following reduction by Steap3, discuss the process through which iron exits endosomes (i.e., describe the role of DMT1 in endosomal iron export). Further phenomenologically, presumably under normal (e.g., without FRDA) conditions, transferrin-transferrin receptor mediated uptake would be controlled by cellular iron status.  If uptake is stimulated, under what conditions would you expect iron released from the endosome to be targeted for storage in ferritin?  Perhaps this is a nuanced discussion, but not all cells have the same cellular demands for iron and this may impact the fate of iron once it is released from the endosome.

OK. We have discussed all these issues as requested (see ll. 163-180)

Line 139: Fe(II) is the preferred iron substrate for ferroportin. As it written, it appears as though the authors are stating that ferrous iron is oxidized prior to export via ferroportin.  If so, what is the oxidase that is involved in the proposed pathway and how does ferroportin export ferric iron?

OK. We adequately discussed this issue to avoid misinterpretations (see ll. 181-187)

Line 140: The review paper referenced discusses the role of both IRP1 and IRP2 (or IREBP2) in the control of ferritin protein synthesis (via translational control). Again, perhaps nuanced, but simply using "expression" does not capture the mode of regulation that is conferred through IRP-dependent mechanisms. Further, the authors' focus should not be solely on IRP2 as this is not reflecting the data that is presented in the reference that is provided.

OK. We have better clarified what is meant for “expression” and reconciled the discussion on IRP2 with the provided reference (see ll. 188-200)

Line 143: The paper by Brewer et al is not appropriately described. Indeed, in this paper, the authors did not inhibit IRP2. Please confirm that this is the reference that was intended to be used.

OK. We apologize for the mistake. We have now changed this reference by citing the work of LaVaute et al (ref. 40) in which an increase of ferritin protein levels co-localizing with high iron content was found in degenerating neurons (see ll. 201-202)

Line 145: The paper that is referenced did not assess iron abundance or availability as a result of HSPB1-mediated control of TfR. Because iron was not actually assessed, please indicate that this is a speculation, but that it is consistent with the model.  It appears as though the authors are over-interpreting the data that was actually provided in the paper that is referenced.

OK. We have clarified that it was a speculation to avoid the overinterpretation of the data provided by the cited paper (see ll. 203-207)

Line 202: Alteration of GPX4 activity? Or expression? Or protein abundance?  Please clarify.

Ok. We have clarified that the alteration was both in terms of mRNA and protein level (see ll. 268) 

Line 203: Revise "Actually, previous studies were focused on other GPXs". As written, it is incomplete. Because of the mention of other GPX isoforms, it would be appropriate to provide additional background related to GPX function and tissue-specific expression. Or, alternatively, focus only on GPX4 (though this would be a mistake).

OK. We have briefly described all the known GPX isoforms (GPX1-8) in terms of function, subcellular localization as well as tissue-specific expression (see ll. 270-289)

Line 211: Rather than reference a review article, the authors should instead reference the original research article from Helveston et al (1996).

OK. We thank the Reviewer for this suggestion and changed the reference accordingly (see new ref. 72)

Line 217: Has the role of GPX4 been "poorly" characterized? This implies that the work has been conducted but it has been of questionable quality.  Consider instead...."the role of GPX4.......FRDA is not well characterized".

OK. We have edited this sentence according to the Reviewer’s suggestion (see line 303). 

Line 229: Revise to remove "a light was shed..."

OK. We have edited this sentence as requested (see ll. 313-314)

Line 236: Consider using "lend" or "provide support" rather than "give effort".

OK. We have edited this term as requested (see Iine 321)

Lines 245-246: It does not appear that the paper that is referenced actually measured mitochondrial iron utilization. Please revise accordingly. Similarly, although Fe-S cluster assembly and/or heme biosynthesis were not assessed. This reviewer did not evidence that Huang et al demonstrated an increase in free iron and/or cellular damage.

OK. We apologize for this mistake and revised the text according to the paper of Huang et al (see ll. 330-339)

Line 283-284: Provide some specific examples of how hypoxia impacts Fe-S cluster biogenesis. Additionally, "...limiting Fenton reactions" is incomplete.

OK. We have provided specific examples on the impact of hypoxia on the biogenesis of Fe-S clusters and edited the discussion on this topic (see ll. 369-375).

Line 285: Again, this can be a more nuanced discussion than is presented, but not all Fe-S containing protein exhibit similar levels of sensitivity to oxygen.

OK. We have mentioned that hypoxia impacts some Fe-S cluster (see line 374) 

Line 299: Remove the hyphen between "demonstrated" and "that".

OK. We have removed it.

Lines 302-305: Clearly describe the impact of the reduced expression of the Xc system.

OK. We have better emphasized the fact that the increase of P53, by repressing SLC7A11 expression, negatively impacts on Xc transporter and lowers intracellular GSH content (see ll. 390-395).

Line 311: Rather than "...very few articles have deeply explored..." consider "the role of ferroptosis in FRDA pathogenesis and disease progress has received relatively little attention" (or little investigation).

OK. We have edited this sentence as suggested (see ll. 401-402).

Round 2

Reviewer 1 Report

The authors have replied to all major and minor comments. I recommend the manuscript for publication.